# Increase in coercive measures in psychiatric hospitals in Germany during the COVID-19 pandemic

**Erich Flammer[1,2], Frank Eisele[2], Sophie Hirsch[1,3], Tilman Steinert**[1,2,4]*

1 Clinic for Psychiatry and Psychotherapy I, Ulm University, Ulm, Germany, 2 Centers for Psychiatry Suedwuerttemberg, Ravensburg, Germany, 3 Centers for Psychiatry Suedwuerttemberg, Biberach, Germany, 4 Department Psychiatry, Tuebingen University, Tuebingen, Germany

* tilman.steinert@zfp-zentrum.de

**Data Availability Statement:** Data are accessible on dryad via https://datadryad.org/stash/share/br-EupCXbygnoCUkVyfLKSDVgqOL86nBxDjgY bqIdF4. The data are anonymized and do not allow identification of patients or hospitals.

## Abstract

### Objective

To examine whether the pandemic in 2020 caused changes in psychiatric hospital cases, the percentage of patients exposed to coercive interventions, and aggressive incidents.

### Methods

We used the case registry for coercive measures of the State of Baden-Wuerttemberg, comprising case-related data on mechanical restraint, seclusion, physical restraint, and forced medication in each of the State's 31 licensed hospitals treating adults, to compare data from 2019 and 2020.

### Results

The number of cases in adult psychiatry decreased by 7.6% from 105,782 to 97,761. The percentage of involuntary cases increased from 12.3 to 14.1%, and the absolute number of coercive measures increased by 4.7% from 26,269 to 27,514. The percentage of cases exposed to any kind of coercive measure increased by 24.6% from 6.5 to 8.1%, and the median cumulative duration per affected case increased by 13.1% from 12.2 to 13.8 hrs, where seclusion increased more than mechanical restraint. The percentage of patients with aggressive incidents, collected in 10 hospitals, remained unchanged.

### Conclusions

While voluntary cases decreased considerably during the pandemic, involuntary cases increased slightly. However, the increased percentage of patients exposed to coercion is not only due to a decreased percentage of voluntary patients, as the duration of coercive measures per case also increased. The changes that indicate deterioration in treatment quality were probably caused by the multitude of measures to manage the pandemic. The focus of attention and internal rules as well have shifted from prevention of coercion to prevention of infection.

**Funding:** The authors received no specific funding for this work.

**Competing interests:** The authors have declared that no competing interests exist.

## Introduction

Coercive measures, particularly involuntary commitment, seclusion, restraint, and forced medication are interventions that deeply violate a patient's autonomy. Such measures should only be used as a last resort, according to the recommendations of international organizations. Recently, a research initiative comprising currently 25 European countries has been established to reduce the use of coercion in mental health services [1]. In Germany, the Federal Constitutional Court decided in 2018 that mechanical restraint is the most restrictive intervention and requires a judge's decision after a personal bedside assessment if lasting longer than 30 minutes [2], which is unique worldwide. At the same time, the German Society for Psychiatry and Psychotherapy (DGPPN) published evidence- and consensus-based guidelines on the prevention of coercion in the treatment of aggressive behavior [3, 4]. In Germany, psychiatric clinics are widely available on a high quality level. About half of these clinics are part of university hospitals or general hospitals, the other half are specialized psychiatric hospitals. Day clinics are widely available, standing alone or as a part of these hospitals. Outpatient treatment is provided by the hospitals for people with severe mental disorders together with community psychiatric services, treatment for milder mental disorders is provided by physicians or psychologists in own practice, funded by health insurances. Community treatment orders or other types of involuntary outpatient commitment are not legalized, so that any type of coercion in association with treatment of mental disorders can only occur within a hospital.

Using the data of the registry for coercive measures in psychiatric hospitals of the State of Baden-Wuerttemberg, we recently demonstrated that the percentage of psychiatric cases that were subjected to restraint or seclusion subsequently decreased by 12%, comparing the years 2017 and 2019, after the decision of the Federal Constitutional Court and the subsequent changes of legislation. Also, the duration of these measures per affected case had decreased by 5% on average [5]. Generally, the topic of coercion was high on the agenda in Germany in recent years, with many awareness workshops and conferences, publications of research groups in German and international journals, funding by research bodies and the German Ministry of Health, and broad implementation of de-escalation trainings [6], and increasing implementation of complex interventions such as the Safewards Model [7].

In this climate of relative open-mindedness and evidence-based strategies to reduce coercion, the COVID-19 pandemic in 2020 affected society and psychiatric hospitals as well, like in all other countries. Surprisingly, mental health in large segments of society did not appear to have deteriorated significantly as a result of the pandemic. In particular, people who were already unwell before the pandemic did not see them deteriorate further in studies. On the other hand, there continue to be warning voices that see a psychiatric pandemic heading toward society.

The pandemic situation imposed specific impacts on psychiatric hospitals: voluntary cases decreased because patients feared infections, former open wards needed to be locked to control the entry of visitors, weekend leave for patients was strictly restricted, visitors were no longer allowed, and group therapies were no longer possible. Hygiene regimes inside hospitals required testing and isolating patients with infections and contact persons as well, and unexpected, sudden staff shortages resulted from infections and quarantine measures [8]. Hence, there were concerns that the use of coercive interventions would increase again, annihilating the achieved improvements in practice. There is evidence from psychiatric hospitals in Germany that this unhappy consequence of the pandemic in fact happened. Fasshauer et al. reported a decrease in the absolute number but an increase in the percentage of emergency hospital admissions in a private hospital group, and the percentage of involuntary admissions increased. The percentage of patients subjected to seclusion or restraint increased compared to

2019, but still remained under the level of 2018 [9, 10]. In contrast, a single hospital in Canada reported a significant decrease in aggression, restraint, and seclusion after the beginning of the COVID-19 pandemic [11]. We could not identify publications from elsewhere on the impact of the pandemic on the use of coercion in psychiatric hospitals at the time.

The Baden-Wuerttemberg registry of coercive measures in psychiatric hospitals [12] enabled us to analyze the changes in the use of coercion after the beginning of the pandemic at the level of a complete Federal State in Germany with 11 million inhabitants. This registry is unique in Germany, as it contains raw data on each coercive measure in all psychiatric hospitals on any legal basis (mental health law, guardianship law, temporary detention). Moreover, ten big hospitals, together serving about half of the population, had introduced a standardized recording of aggressive incidents some years ago, so that data on aggressive behavior are also available. The objective of this study was to analyze changes in cases, involuntary cases, seclusion, restraint, coercive medication, and aggressive incidents in the first year of the pandemic (2020) compared to the year before and see if coercive measures and aggressive incidents increased during the pandemic.

## Methods

### Coercive incidents: Data sources

In 2015, a new mental health law was introduced in the German federal state of Baden-Wuerttemberg following a Constitutional Court decision. It contained the unique feature of requiring all 32 public psychiatric hospitals to collect data on seclusion, restraint, and forced medication in emergency situations or by judicial order. Raw data on each coercive measure in all hospitals are reported to the registry. This procedure has special requirements for data protection and data security considering highly sensitive personal data. An online platform was set up after detailed consultation with the state data privacy and data security officer and his final approval. The platform serves for both uploading data by the institutions and downloading data by the evaluation office. Data privacy is ascertained by a double and irreversible pseudonymization carried out by different institutions and through the use of passwords. Thus, the identification of individual persons is not possible, i.e., the data are anonymized.

For each coercive intervention, the dataset contains the kind of intervention as defined by a codebook, its legal basis, the duration, the patient's gender, the ICD-10 principal group, and a pseudonymized case ID. This allows assigning coercive measures with identical pseudonymized case numbers to the same case, which is necessary to determine the outcomes according to the study questions. Because the occurrence of coercive incidents can only be determined after a patient has been discharged, cases are defined as discharges in a reporting year, irrespective whether the case occurred in the previous or in the current reporting year. For this reason, we use the term "case" (and not the term "admission", though the figures would be roughly identical). While the registry contains raw data on coercive measures (not on the numbers of cases) it does not contain information whether two or more cases represent the same patient across different cases. For all hospitals, the number of cases with respect to diagnoses and the number of involuntary cases according to different laws are available [12]. The numbers of cases according to diagnoses and involuntary cases, based on public law or guardianship law, are reported as cumulative numbers by the hospitals.

Hospitals must deliver data for the previous year before a deadline. The data are then checked for completeness and plausibility. In case of abnormalities, the clinics concerned are consulted and if necessary and possible, the data is corrected. After this data check, descriptive evaluations are carried out. The results of these evaluations are reported to the hospitals and to the Ministry of Social Welfare and Integration of Baden-Wuerttemberg in a standardized

annual report. Once in the legislative period, a report to the state parliament of Baden-Wuert-temberg is made by the Ministry of Social Welfare and Integration. Further details have been reported elsewhere [12].

## Aggressive incidents: Data sources

The Staff Observation Aggression Scale–Revised (SOAS-R) was introduced for regular use and reporting in 10 out of the 31 hospitals within the last decade. The SOAS-R is a one-page form that can be filled within a few minutes by staff without training after an aggressive incident. It comprises five domains: provocation (score 0–2), means used by the patient (score 0–3), target of aggression (score 0–4), consequence(s) for victim(s) (score 0–9), and measure(s) to stop aggression (score 0–4). The possible range of the total score results in 1–22 points. The scale has been extensively used in research on violent incidents in psychiatric hospitals in Europe in the past 30 years. Characteristics of the scale and methods of recording have been reported in detail elsewhere [13–15]. Due to some doubts with respect to fully covering self-directed aggression, we restrict the analysis to aggression toward others and toward objects as indicated in domain 2, target of aggression.

## Ethics

The Ethics Committee of Ulm University waived the requirement for ethical approval as approval is not required for studies analyzing anonymized data, in accordance with national legislation and institutional requirements.

## Definitions

Definitions of coercive measures and detailed prescriptions for recording them with respect to duration and legal grounds are available in a codebook provided for the hospitals by the Ministry of Health, Social Welfare, and Integration. There have been only very minor changes since 2015. All use of freedom-restricting devices has to be recorded as mechanical restraint, encompassing not only belts in beds, but also (undivided) bedrails, movement-restricting blankets, tables attached to a chair, and other devices in old age psychiatry. Physical restraint (staff holding a person for a period of time by force) is rare in psychiatry in Germany [6], but is recorded separately. Seclusion is defined according to suggestions in the literature [3] as locking a person in a scarcely furnished room (mostly with only a mattress and toilet) without the presence of staff. Chemical restraint is uncommon as a category in Germany. Forced medication can be administered only in cases of acute emergency or for therapeutic reasons after an independent expert review and a judge's decision. Based on these legal prerequisites, involuntary medication was classified as either emergency medication or medication according to a court decision.

## Study design

We used an observational prospective design and compared data from Baden-Wuerttemberg's 31 licensed hospitals treating adults (one hospital is only licensed for child- and adolescent psychiatry and therefore was excluded from this analysis) on coercive measures, forced medication, and aggressive incidents in adult psychiatry from 2019 (before the pandemic) with data from the first year of the pandemic (2020). Forensic psychiatry is also part of the registry, but was not included in this analysis. Hence, this is a full survey, comprising each coercive measure that had been performed in one of the State's psychiatric hospitals in the respective year in adults. Baden-Wuerttemberg with 11 million inhabitants covers the South West of Germany,

with Stuttgart as the State's capital. Data are available for the respective years, but, due to data privacy rules, the exact date of incidents is not provided so that we could not restrict our analysis to the months of the pandemic (beginning in March, 2020). This may have led to a systematic underestimation of observed changes of about 15%.

In addition, we could collect aggressive incidents as recorded with the SOAS-R from 10 of the 31 hospitals. These ten hospitals belong to a State-run company. Most of them are divided into several sites, and together they serve about half of the population. The data structure and ways of data collecting were very similar as described for coercive measures. However, the dataset does not allow for relating reports on aggressive incidents and coercive measures on individual patient level.

## Outcomes

In line with previous work with similar methods [15, 16], we chose seven outcomes, (1) the percentage of cases on any involuntary legal basis, (2) the percentage of cases that were affected by mechanical restraint, seclusion, physical restraint, emergency medication, or forced medication, (3) the duration of seclusion, mechanical, or physical restraint episodes, (4) the cumulative duration of seclusion, mechanical, or physical restraint per affected case, (5) the percentage of cases in whom aggressive behavior towards others was recorded by the SOAS-R, (6) the SOAS-R score, and (7) the number of aggressive incidents with injurious consequences.

## Analyses

To assess the impact of the COVID-19 pandemic, we analyzed changes in the number of treated cases, involuntary cases, seclusion, restraint, coercive medication, and aggressive incidents between 2019 and 2020. Therefore we compared the percentage of affected cases and the median (inter-quartile range, IQR) duration of coercive measures and the cumulative duration of coercive measures per affected case for 2019 with the respective data for the year 2020.

To assess the statistical significance of differences we used the chi-squared test for the proportion of affected cases and the Mann-Whitney U test for the duration of coercive measures. We chose the Mann-Whitney U test as the data were heavily skewed. For the SOAS-R score, we used t-test for independent samples. We also calculated effect sizes. For the differences in the proportions of cases with coercive measures, we calculated unadjusted risk ratios (RR) and 95% confidence intervals (95%-CI), and for the differences in the median cumulated duration of coercive measures and for the difference in the SOAS-R score, we calculated effect sizes eta squared and converted them into common language effect sizes Cohen's d [17, 18]. Analyses were done with IBM® SPSS® Statistics Version 27, Microsoft® Excel® 2013. Effect sizes for duration of coercive measures and for the difference in the SOAS-R score were calculated online [18].

## Results

### Involuntary cases and coercive measures

From 2019 to 2020, the number of cases in adult psychiatry decreased by 7.6% from 105,782 to 97,761, while the absolute number of involuntary cases increased slightly and the percentage of all cases increased from 12.3% to 14.1% (p < .001). This increase was similar for all legal procedures, i.e. caring detention (patients forced to stay in the hospital by a physician before a court's decision), involuntary cases according to civil law, and involuntary cases according to public law (Table 1). The percentage of cases exposed to any kind of coercive measure

increased by 24.6% from 6.5% in 2019 to 8.1% in 2020 (p < .001). This effect was largest for seclusion (Table 1). The percentage of involuntary cases varied between hospitals from 0.2% to 26.0% in 2019 and from 0.3% to 34.3% in 2020. The percentage of cases subjected to any kind of coercive measure varied between hospitals from 0.1% to 10.3% in 2019 and from 0.2% to 12.7% in 2020.

The absolute number of coercive measures increased by 4.7% from 26,269 in 2019 to 27,514 in 2020 (Table 2). The median duration of mechanical restraint, seclusion or physical restraint episodes increased by 11.1% from 6.3 hours to 7.0 hours (p < .001). When looking at these coercive measures individually, only the median duration of seclusion increased statistically significantly (Table 2).

From 2019 to 2020, the median cumulative duration of mechanical restraint, seclusion or physical restraint episodes per affected case increased by 13.1% from 12.2 hours to 13.8 hours

**Table 1. Cases and percentages exposed to coercive interventions in 2020 compared to 2019.**

|  | 2019 | 2020 | p-value |
|---|---|---|---|
|  |  |  | Effect size [95%- CI] |
| **Number of cases** | 105,782 | 97,761 |  |
| **Number of involuntary cases (%)** | 13,032 | 13,824 | p < .001 |
|  | (12.3%) | (14.1%) | RR = 1.15 |
|  |  |  | [1.12; 1.17] |
| **Number of cases with caring detention (%)** | 6,138 | 6,357 | p < .001 |
|  | (5.8%) | (6.5%) | RR = 1.12 |
|  |  |  | [1.08; 1.16] |
| **Number of involuntary cases according to civil law (%)** | 3,321 | 3,590 | p < .001 |
|  | (3.1%) | (3.7%) | RR = 1.17 |
|  |  |  | [1.12; 1.23] |
| **Number of involuntary cases according to public law (%)** | 3,573 | 3,877 | p < .001 |
|  | (3.4%) | (4.0%) | RR = 1.17 |
|  |  |  | [1.12; 1.23] |
| **Number of cases subjected to any kind of coercive measures (%)** | 6,853 | 7,912 | p < .001 |
|  | (6.5%) | (8.1%) | RR = 1.25 |
|  |  |  | [1.21; 1.29] |
| **Number of cases subjected to mechanical restraint (%)** | 4.087 | 4,134 | p < .001 |
|  | (3.9%) | (4.2%) | RR = 1.09 |
|  |  |  | [1.05; 1.14] |
| **Number of cases subjected to seclusion (%)** | 3,807 | 4,989 | p < .001 |
|  | (3.6%) | (5.1%) | RR = 1.42 |
|  |  |  | [1,36: 1,48] |
| **Number of cases subjected to physical restraint (%)** | 100 | 94 | n.s.* |
|  | (0.1%) | (0.1%) | RR = 1.02 |
|  |  |  | [0.77; 1.35] |
| **Number of cases subjected to emergency or forced medication (%)** | 907 | 946 | p < .01 |
|  | (0.9%) | (1.0%) | RR = 1.13 |
|  |  |  | [1.03; 1.24] |
| **Number of cases subjected to coercive measures not specified (%)** | 55 | 45 | n.s.* |
|  | (0.1%) | (0.0%) | RR = 0.89 |
|  |  |  | [0,60; 1,31] |

*n.s.: not significant

**Table 2. Number and duration of coercive episodes in 2020 compared to 2019.**

|  | 2019 | 2020 | p-value |
|---|---|---|---|
|  |  |  | Effect size [95%-CI] |
| **Number of coercive episodes of any kind (per treated case)** | 26,269 | 27,514 | p < .001 |
|  | (0.25) | (0.28) | RR = 1.13 |
|  |  |  | [1.12; 1.15] |
| **Number of mechanical restraint episodes (per treated case)** | 10,486 | 9,188 | p < .001 |
|  | (0.10) | (0.09) | RR = 0.95 |
|  |  |  | [0.92; 0.97] |
| **Number of seclusion episodes (per treated case)** | 13,730 | 15,897 | p < .001 |
|  | (0.13) | (0.16) | RR = 1.25 |
|  |  |  | [1.23; 1.28] |
| **Number of physical restraint episodes (per treated case)** | 132 | 94 | n.s.* |
|  | (0.0012) | (0.0010) | RR = 0.77 |
|  |  |  | [0.59; 1.00] |
| **Number of emergency or forced medications (per treated case)** | 1,758 | 1,774 | p < .05 |
|  | (0.017) | (0.019) | RR = 1.09 |
|  |  |  | [1.01; 1.15] |
| **Number of coercive measures not specified (per treated case)** | 163 | 488 | p < .001 |
|  | (0.002) | (0.005) | RR = 3.24 |
|  |  |  | [2.71; 3.87] |
| **Duration of mechanical restraint, seclusion or physical restraint episodes (median (hrs), [IQR])** | 6.3 | 7.0 | p < .001 |
|  | [2.0; 14.9] | [2.0; 16.8] | d = 0.05 |
| **Duration of mechanical restraint episodes (median (hrs), [IQR])** | 5.8 | 5.8 | n.s.* |
|  | [2.0; 13.0] | [1.8; 13.6] | d = 0.007 |
| **Duration of seclusion episodes (median (hrs), [IQR])** | 7.1 | 8.0 | p < .001 |
|  | [2.3; 16.8] | [2.3; 18.9] | d = 0.08 |
| **Duration of physical restraint episodes (median (hrs), [IQR])** | 0.2 | 0.2 | n.s.* |
|  | [0.1; 0.4] | [0.1; 0.3] | d = 0.14 |

*n.s.: not significant

(p < .001). When considered separately, only the median cumulative duration of seclusion increased statistically significantly (Table 3).

## Aggressive Incidents

The number of discharged cases of the 10 hospitals that have implemented the SOAS-R as a reporting system decreased by 7.6% from 60,484 to 55,863, while the number of discharged cases with aggressive incidents remained almost unchanged. As a result, the proportion of cases with aggressive incidents increased from 7.5% to 8.0% (Table 4). The percentage of cases with aggressive incidents varied between hospitals from 3.9% to 8.8% in 2019 and from 4.1% to 10.1% in 2020. The mean SOAS-R score varied between hospitals from 10.0 to 14.1 in 2019 and from 10.2 to 13.7 in 2020. The percentage of aggressive incidents with injury consequences varied between hospitals from 14.3% to 50.3% in 2019 and from 12.7% to 44.1% in 2020.

Similarly, the total number of aggressive incidents remained roughly constant, with 15,657 in 2019 and 15,669 in 2020. The mean SOAS-R score also changed only slightly, rising from

Table 3. Cumulated duration of coercive episodes per affected case in 2020 compared to 2019.

| | 2019 | 2020 | p-value |
| --- | --- | --- | --- |
| | | | Effect size |
| Median cumulated duration (hrs) of mechanical restraint, seclusion or physical restraint episodes per affected case [IQR] | 12.2 | 13.8 | p < .001 |
| | [4.3; 32.5] | [4.7; 38.4] | d = 0.06 |
| Median cumulated duration (hrs) of mechanical restraint episodes per affected case [IQR] | 8.8 | 8.5 | n.s.* |
| | [2.8; 24.5] | [2.5; 23.4] | d = 0.005 |
| Median cumulated duration (hrs) of seclusion episodes per affected case [IQR] | 12.0 | 14.2 | p < .001 |
| | [4.3; 29.3] | [4.9; 37.4] | d = 0.10 |
| Median cumulated duration (hrs) of physical restraint episodes per affected case [IQR] | 0.3 | 0.3 | n.s.* |
| | [0.2; 0.8] | [0.1; 0.6] | d = 0.13 |

*n.s.: not significant

11.9 to 12.1. The proportion of aggressive incidents with injury consequences also remained unchanged (Table 5).

## Discussion

In 2020, Wilson [19] described the possible detrimental effects of the pandemic on the legal position and the human rights of people with mental illnesses, particularly on all aspects of involuntary cases and treatment. The considerations outlined there for Australia are probably valid for all high income countries. She expressed her concerns that there are no publicly available data on the impact of the pandemic on this vulnerable population. Now we can present such data based on a total survey of all coercive interventions in psychiatric hospitals in a Federal State with 11 million inhabitants, encompassing over 200,000 cases in the years 2019 and 2020. The comparison of the data in the year before the pandemic (2019) and the first year of the pandemic (2020) confirms the devastating effects of the COVID-19 pandemic on previous achievements to reduce coercion in psychiatry, as demonstrated in the same population [5, 16]. The number of hospital cases decreased considerably; involuntary cases, however, increased slightly and consequently their proportion of all cases increased. This is in line with other recent publications from Germany [9, 10, 20]. The same applies for the absolute number of coercive measures and, additionally, the percentage of cases exposed to freedom-restrictive coercive measures increased by nearly 25%. This result is even conservative since due to methodological limitations we could compare only data sets from complete calendar years and the

Table 4. Cases and cases with aggressive incidents in 2020 compared to 2019.

| | 2019 | 2020 | p-value |
| --- | --- | --- | --- |
| | | | Effect size [95%- CI] |
| Number of cases | 60,484 | 55,863 | |
| Number of cases with aggressive incidents (%) | 4,564 | 4,452 | p < .01 |
| | (7.5%) | (8.0%) | RR = 1.06 |
| | | | [1.02; 1.10] |

*n.s.: not significant

**Table 5. Number of aggressive incidents in 2020 compared to 2019.**

| | 2019 | 2020 | p-value |
|---|---|---|---|
| | | | Effect size [95%- CI] |
| **Number of aggressive incidents** | 15,657 | 15,669 | |
| **Mean SOAS-R score (SD)** | 11.9 | 12.1 | p < .001 |
| | (4.9) | (4.7) | d = 0.042 |
| **Number of aggressive incidents with injury consequences (%)** | 3,813 | 3,814 | n.s.* |
| | (24.4%) | (24.3%) | RR = 1.0 |
| | | | [0.96; 1.04] |

impact of the pandemic enfolded only during the course of March, 2020. The data suggest that the most severely ill patients continued to receive care, if necessary, on an involuntary basis, while less severely ill patients tended to avoid hospital care themselves or were not admitted due to the very restrictive case policy of hospitals. A similar development was observed in most medical specialties [9, 10] where it was also followed by a decline of treatment quality, especially a delay of treatment as well as a higher burden of disease.

Notably, cases of severe mental disorders directly caused by an infection with COVID-19 or anxiety associated with the pandemic, vaccinations or lockdown measures were anecdotally reported [21] but had little impact on the number of in-patient admissions. The frequently observed depressive and anxiety disorders were mostly treated in out-patient services, also by video consultations [22] and in newly established services for this purpose in Baden-Wuerttemberg [23].

Notwithstanding the fact that the presented longitudinal observational data do not allow for causal inferences in their nature, with respect to the use of coercion, we are not aware of any other explanation for this State-wide phenomenon. Moreover, the calculations are rather conservative and may even underestimate the effects, since the impact of COVID-19 on daily life in Germany occurred in March 2020. Due to data privacy regulations, we cannot separate the first two months of 2020 from the rest of the year in the analyses. The increase in seclusion and the parallel reduction in mechanical restraint are probably not due to effects of the pandemic, but reflect a trend that had already been observed previously, following legal regulations [16].

Our data does not allow inferences on the reasons for the increase in coercion in psychiatric hospitals in detail. However, there is plenty of at least anecdotal evidence from conferences and a limited number of publications [8, 10]. Notably, the number of psychiatric patients with COVID-19 infection remained small throughout the year (and is not known exactly), and isolation due to regulations of hygiene and disease control certainly accounted only for a relatively small percentage of seclusion and restraint measures. If possible, infected patients were not admitted, discharged, or transferred to somatic hospitals in cases of severe disease. Nevertheless, considerable outbreaks among patients and staff and difficult-to-manage situations occurred repeatedly and required the establishment of isolation units and their continuous staffing. However, clinicians argue that the observed increase in coercion was caused much more by the indirect effects of the pandemic than by patients infected with COVID-19 themselves. There is a bundle of resulting adverse circumstances; part of it has been described by Gather et al. [8]. Open door policies were abandoned not because of the danger of absconding, but to prevent uncontrolled visitors from introducing infections. For the same reason, weekend leaves and unaccompanied leaves from wards were restricted, and group therapies (psychotherapy, occupational therapy, arts therapy, and sports therapy as well) were no longer feasible. Communication was generally complicated by the requirement to wear face masks.

Generally, continuous trustful relationships with patients are hampered if staff persons fall ill or go into quarantine and have to be replaced by staff from other wards in the short term. Remaining staff were considerably occupied by tasks such as testing themselves, patients, and visitors, and discussions on hygiene measures and necessary documentation requirements. Educational programs, for instance in de-escalation, can be sustained only to a limited extent, e.g. by online teaching. The focus of attention has necessarily shifted from the prevention of coercion to prevention of infection.

Our study has the typical limitations of observational studies. Even if it might look rather obvious in the present case, conclusions referring to causal attributions remain speculative and are not supported by data. Another limitation is the very likely presence of unknown confounders. In a previous study on differences between hospitals in the rates of coercive measures, only 27% of the variation could be explained by a wide range of structural data of hospitals and supply areas [24]. With respect to data quality, no findings are available on the reliability of the data collected. It may also be that, despite detailed instructions, there is not sufficiently uniform recording across clinics. So underreporting or incorrect reporting cannot be ruled out. Beyond the presented empirical data, no systematic knowledge is available on the consequences of the COVID-19 pandemic on everyday clinical practice in psychiatric hospitals. Further in-depth qualitative research will be necessary for a deeper understanding of the detrimental consequences of the pandemic situation on different patient groups in psychiatric hospitals, day clinics, and outpatient and rehabilitation services.

## Conclusions

The COVID-19 pandemic had detrimental effects on the achievements to avoid coercion during in-patient psychiatric treatment. Necessarily, the focus of attention and rules as well shifted from prevention of coercion to prevention of infections. By and large, this was not due to the admission of patients with COVID-19 or patients with COVID-19-related disorders, but due to the many adverse circumstances of the pandemic for the conditions of humane and respectful in-patient treatment.

## Author Contributions

**Conceptualization:** Tilman Steinert.

**Data curation:** Frank Eisele.

**Formal analysis:** Erich Flammer, Frank Eisele.

**Methodology:** Erich Flammer, Tilman Steinert.

**Validation:** Sophie Hirsch.

**Writing – original draft:** Tilman Steinert.

**Writing – review & editing:** Erich Flammer, Sophie Hirsch, Tilman Steinert.

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
