## [Decision Letter · Decision Letter 0]

1 May 2022

PONE-D-22-03053Increase in coercive measures in psychiatric hospitals in Germany during the COVID-19 PandemicPLOS ONE

Dear Dr. Steinert,

Thank you for submitting your manuscript to PLOS ONE. After careful consideration, we feel that it has merit but does not fully meet PLOS ONE’s publication criteria as it currently stands. Therefore, we invite you to submit a revised version of the manuscript that addresses the points raised during the review process.

We look forward to receiving your revised manuscript.

Kind regards,

Anshuman Mishra, PhD

Academic Editor

PLOS ONE

Journal Requirements:

Additional Editor Comments (if provided):

Dear Author, Study entitled-Increase in coercive measures in psychiatric hospitals in Germany during the COVID19 Pandemic by Tilman Steinert et al 2022. Study examine changes in psychiatric hospital cases with effect of pandemic regarding the percentage of patients exposed to coercive interventions, and aggressive incidents. Study shows that increased percentage of patients exposed to coercion is not only due to a decreased percentage of voluntary patients, as the duration of coercive measures per case also increased. The focus of attention has shifted from prevention of coercion to prevention of infection.

Article is excellent for the psychiatric perspective, presented in an intelligible fashion and written in standard english however, technical and statistical explanation required extensively for the better understanding of the article. I had appended the reviewer comment for the better understanding and to make article in better shape.

Along with the reviewer comment following suggestions are also given below for the article.

1. Understanding working culture of people, varied clinical complications and regional risk is important for the study (Mental effect of covid-19 pandemic on affected, non affected and families of the affected. Gupta, A 2020. International Journal of Research in Pharmaceutical Sciences. 11(Special Issue 1), pp. 1804-1808.

2. Study of the psychiatric through behavioral changes is also important to understand the risks (COVID-19: Risk of alcohol abuse and psychiatric disorders. Haddadi, S., Murthi, M., Salloum, I., Mirsaeidi, M.S. 2020. Respiratory Medicine Case Reports. 31,101222.

3. Advanced treatment pattern, suggestions, technologies and future prospects of the study will add on more to make the article in better shape.

(Telephone information service for psychiatric patients during the covid-19 pandemic: Experience with a direct phone line in the nyírő gyula national institute of psychiatry and addictions in hungary | [Beteginformációs telefonvonal a covid-19-járvány idején: A nyírő gyula-opai-ban működtetett közvetlen vonallal szerzett tapasztalataink]. Katalin, C., Viktor, B., Krisztina, L., (...), Tünde, V., Szabolcs, K. 2020. Neuropsychopharmacologia Hungarica. 22(4), pp. 166-171. And Virtual reality exposure therapy (Vret) for anxiety due to fear of covid-19 infection: A case series. Zhang, W., Paudel, D., Shi, R., (...), Zhou, Y., Zhang, B. 2020. Neuropsychiatric Disease and Treatment. 16, pp. 2669-2675).

Decision- Major revision with response to all reviewers comment pointwise.

Reviewers' comments:

Reviewer's Responses to Questions

**Comments to the Author**

1. Is the manuscript technically sound, and do the data support the conclusions?

Reviewer #1: Partly

Reviewer #2: Partly

Reviewer #3: No

2. Has the statistical analysis been performed appropriately and rigorously? 

Reviewer #1: No

Reviewer #2: No

Reviewer #3: No

3. Have the authors made all data underlying the findings in their manuscript fully available?

Reviewer #1: No

Reviewer #2: Yes

Reviewer #3: No

4. Is the manuscript presented in an intelligible fashion and written in standard English?

Reviewer #1: Yes

Reviewer #2: Yes

Reviewer #3: Yes

5. Review Comments to the Author

Reviewer #1: Although Work is good and well But there are some points, after improvment of those paper can be acceptable.

1.Explain the method of sample size determination

2. Define clearly different type of samples.

3. Result and statistical analysis is not properly clear.

4. Discussion is not clearly written

Reviewer #2: Authors look at the topic of increases in coercive measures during COVID-19 and find use of some practices increased while others decreased as a trade-off in treatment quality. The contrast in practices was interesting to read in the abstract. Possible reasons for this change could be expanded upon in some way.

Authors provide research from their work showing trends have decreased slightly over time, so this seems like an important follow-up study to consider the impact of the pandemic on rates. While there isn’t much literature on rates that has been published, it may help if authors could discuss a bit more broadly on the impact of the pandemic on care quality to help frame or present a stronger rationale for the expected change in treatment quality rates.

Authors discuss important points and limitations in using the data that make it hard to do research in this area given reporting practices which is helpful to address upfront. I’ve had similar issues with getting “real” values at the patient or incident level as well (and around the right time frame), so appreciate the description.

Authors mention the SOAS-R measure in the Methods. It was unclear if or how this fit within the context of the study. Up to that point, the focus seemed to be on coercive measures (although now re-reading the abstract it is a bit clearer). Further background and description of this measure and its consideration in the study would be helpful

if retaining in the manuscript.

In the study design section, authors should note having data on SOAS-R from 10 hospitals (mentioned earlier in the methods).

Analysis plan seems fairly straight forward. It would help if some additional considerations were part of this approach. It seemed like authors had collected more data about the characteristics of patients at each site. It would help to discuss whether patterns were observed consistently across hospitals or if some places saw different patterns than others.

Table 2 – some of the p-values appear to be missing.

A stronger analysis plan would help strengthen the study’s overall impact and interpretation of findings.

Manuscript would be strengthened by having an overall conclusion paragraph rather than focusing on limitations.

Reviewer #3: Thank you for asking me to review this paper.The topic area is important relevant and interesting, the data registry upon which it is based a possible strength. My main observation of this work however is that the statistical analyses as they’re currently presented seem to indicate that the analysis was mainly descriptive, i.e. not taking approaches that could account for confounding (relative risks are mentioned but it does not appear that any regression approaches were used in the analysis). This is a major limitation and at the very least needs to be reflected upon as a concern in the discussion, however it would be preferable if the analytic methods could go further to address this in the manuscript.

Some other comments -

1. Abstract- some context on country would be helpful for an international readership. Perhaps add “a region in Germany” after ‘State of Baden-Wuerttemberg’

2. The last statement in the abstract ‘The focus of attention has shifted from prevention of coercion to prevention of infection.’ is a little unclear and goes further than what the results suggest

Manuscript in general

1. Some typos - manuscript should be proofed carefully for these eg. ‘leaves’ should be leave pg4, under ‘ Study design“ missing the word ‘from’ , etc. Please re-read and check throughout for small grammatical and spelling errors.

Methods

1. More detail would be helpful- does the registry represent all hospitals in the region , if not which are not represented (proportions?). In Germany would all secondary mental healthcare be provided by these institutions or can people receive secondary care from outside of these institutions?

2. Some mention is made of “Staff Observation Aggression Scale – Revised (SOAS-R) “. Although some details are provided with references further detail would be helpful, rather than expecting readers to have to look up the other papers.It would be helpful to know which domains are covered in this scale, who fills out the scale and relevant psychometric properties of the scale. Mean scores are presented later but these are meaningless to readers unfamiliar with this scale.

3. This line needs further explanation “Due to data privacy rules, the exact date of incidents was not available so that we could not restrict our analysis to the months of the pandemic (beginning in March, 2020). This may have led to a systematic underestimation of observed changes of about 15%.” If you could not restrict analysis to the month of the pandemic how are you able to determine which events occurred prior to the pandemic (2019) versus During the pandemic (2020)? How did you determine an underestimation percentage of 15%? Please explain this in more detail.

4. Statistical methods- more detail needed- what package did you use (spss/ stata etc)? How were the relative risks calculated? Presumably these are crude or unadjusted? If these relative risks were calculated through regression models Did you consider adjusting for confounders? Also did you check underlying assumptions for using Cohen’s D were not violated - cohens D has similar assumptions to T tests

5. Tables: rather than RR with P values It would be preferable to have 95% confidence intervals for the RR.This would allow us to see the strength of Association alongside precision.

6. Table 2- first few rows - it is difficult to assess if these absolute differences are meaningful without a denominator

7. In the tables which statistical tests do the p values relate to? A footnote would be helpful . Rather than showing p=0.000 -standard practice would be to show a boundary eg p<0.001 etc or to two decimal places.

8. Discussion - this detail “Now we can present such data based on a total survey of all coercive interventions in psychiatric hospitals in a Federal State with 11 million inhabitants, encompassing over 200,000 cases in the years 2019 and 2020.” Or something equivalent should be presented earlier on eg in methods.

9. The discussion section could State more about the limitations of the study. There are several obvious ones such as confounding which should be mentioned, in addition there are others which the authors could specify

10. Follow STROBE guidelines in your reporting

6. PLOS authors have the option to publish the peer review history of their article (what does this mean?). If published, this will include your full peer review and any attached files.

Reviewer #1: **Yes: **Dr. Medhavi Sudarshan

Reviewer #2: No

Reviewer #3: No

---

## [Author Response · Author response to Decision Letter 0]

3 Jun 2022

we provide an extensive response letter enclosed.

---

## [Editor Report · Decision Letter 1]

9 Jun 2022

Increase in coercive measures in psychiatric hospitals in Germany during the COVID-19 Pandemic

PONE-D-22-03053R1

Dear Dr. Steinert,

We’re pleased to inform you that your manuscript has been judged scientifically suitable for publication and will be formally accepted for publication once it meets all outstanding technical requirements.

Kind regards,

Anshuman Mishra, PhD

Academic Editor

PLOS ONE
---

## [Editor Report · Acceptance letter]

10 Aug 2022

PONE-D-22-03053R1 

Increase in coercive measures in psychiatric hospitals in Germany during the COVID-19 Pandemic 

Dear Dr. Steinert:

I'm pleased to inform you that your manuscript has been deemed suitable for publication in PLOS ONE. Congratulations! Your manuscript is now with our production department. 

Kind regards, 

on behalf of

Dr. Anshuman Mishra 

Academic Editor

PLOS ONE